# Properties of WC–10%Co–4%Cr Detonation Spray Coating Deposited on the Al–4%Cu–1%Mg Alloy

**DOI:** 10.3390/ma14051206

**Published:** 2021-03-04

**Authors:** Marina Samodurova, Nataliya Shaburova, Olga Samoilova, Ahmad Ostovari Moghaddam, Kirill Pashkeev, Vladimir Ul’yanitckiy, Evgeny Trofimov

**Affiliations:** 1Resource Center for Special Metallurgy, South Ural State University, 76 Lenin Av., Chelyabinsk 454080, Russia; samodurovamn@susu.ru (M.S.); pashkeevki@susu.ru (K.P.); 2Department of Materials Science, Physical and Chemical Properties of Materials, South Ural State University, 76 Lenin Av., Chelyabinsk 454080, Russia; samoylova_o@mail.ru (O.S.); ostovary@aut.ac.ir (A.O.M.); tea7510@gmail.com (E.T.); 3Laboratory of Synthesis of Composite Materials, Institute of Hydrodynamics, M.A. Lavrent’ev SB RAS, 15 Lavrent’ev Av., Novosibirsk 630090, Russia; ulianv@mail.ru

**Keywords:** detonation spray, WC–Co–Cr carbide coating, microstructure, hardness, cohesive strength

## Abstract

One of the methods of local improvement of the wear resistance of aluminum alloy parts is the deposition of hard tungsten carbide-based coatings on the surfaces subjected to intense external influence. This paper is devoted to the characterization of the WC–10Co–4Cr (wt.%) coating deposited on an Al–4Cu–1Mg (wt.%) alloy by the detonation spray method. In comparison with the common thermal spray techniques like High Velocity Oxygen Fuel (HVOF) or Atmospheric Plasma Spraying (APS), the heat input delivered to the substrate during detonation spray is significantly lower, that is especially important in case of coating deposition on aluminum alloys. The paper presents the results of morphology investigation, microstructure, phase composition, microhardness, and cohesive strength of deposited carbide-based detonation spray coating. Results showed that the coating has a porosity less than 0.5% and the carbide grain refinement down to the submicron size during coating deposition was detected. According to the investigation, the variation of spraying distance from 270 to 230 mm does not influence on the coating microstructure and composition.

## 1. Introduction

Aluminum and its alloys are traditionally classified as materials with low hardness and low tribological characteristics. In particular, low abrasive wear resistance during friction was reported by Udoye et al. [1]. These properties of aluminum alloys can be enhanced by surface hardening performed by micro-arc oxidation (MAO), plasma electrolytic oxidation (PEO) and deposition of hard materials by electroplating [2,3,4,5,6,7,8,9,10,11,12,13], or by thermal spraying of hard coatings based on tungsten, titanium, and tantalum carbides with metal bond. The most common method for the thermal spraying of WC-based coatings applied in industry is the High Velocity Oxygen Fuel (HVOF) process [14,15,16,17,18,19,20,21]. Another thermal spray method capable to produce hard cermet coatings is the detonation spray [22,23,24,25,26,27,28,29,30,31]. According to Nenashev et al. [27] and Ulianitsky et al. [31], the carbide-based cermet coatings produced by the detonation method has fine structure with sub-micron tungsten carbide inclusions that increases the wear-resistance of the coating. The refinement of the structure of the WC–Co detonation coating in comparison with plasma spraying was also reported in the work Roy et al. [28]. Authors [28] also noted that detonation spray coating is also characterized by increased hardness and abrasion resistance, in comparison to the atmospheric plasma spray deposits. Another important advantage of detonation spray is the limited heat input to the substrate, that is particularly important in the case of deposition on aluminum alloy parts. In contrast to the HVOF spray with continuous flame torch, in detonation spray the particles are accelerated and heated by short explosions. The control of time interval between explosions allows to adjust the heat transfer to the substrate in order to avoid the part deterioration due to thermal stresses. The properties of detonation spray carbide-based coatings directly applied to an aluminum surface were discussed in [29,30,31]. In the work by Roy [29], the results of dynamic hardness measurements of WC–Co coatings on mild steel and aluminum substrates were presented. The measurements showed that the coatings deposited on aluminum substrates had higher dynamic hardness than in case of mild steel, because the porosity of the coating on aluminum turned out to be lower than on steel. In the work by Roy [30], the author showed that the characteristics of the erosion resistance of the WC–12 (wt.%) Co detonation coating are not inferior to those of the same coating applied by the electrospark deposition. According to the Ulianitsky et al. [31], the WC–25 (wt.%) Co coating deposited on aluminum had high adhesion strength, comparable with the cohesion strength of the coating.

It is important to note that the results presented in literature described mainly the properties of tungsten carbide coatings with the WC phase percentage in feedstock powder not higher than 75%. In case of deposition of cermet powders with higher carbide percentage (with WC content up to 86–88%) significantly higher energy explosion energy should be used in order to obtain dense and uniform coating. However, an increase in the explosion energy could lead to undesirable effects related to the overheating of aluminum surfaces, with the formation of new phases on the coating-substrate interface. Unfortunately, the data presented in literature do not fully describe the properties of detonation spray coatings with WC content up to 86–88% deposited on the aluminum alloy substrate.

In the other hand, in comparison with the common thermal spray techniques like High Velocity Oxygen Fuel (HVOF) or Atmospheric Plasma Spraying (APS), the heat input delivered to the substrate during detonation spray is significantly lower, that is especially important in case of coating deposition on aluminum alloys.

In our opinion, the application of high-carbide coatings (with WC content up to 86–88%) on aluminum alloys by the detonation method should not lead to a deterioration in the quality of the resulting coating, which we intend to show in our study.

The aim of this work was to analyze the WC–10Co–4Cr (wt.%) coating morphology, as well as the microstructural, mechanical, and cohesive properties of a detonation spray coating deposited on an Al–4Cu–1Mg (wt.%) alloy.

## 2. Materials and Methods

Aluminum alloy (LLK KMK Steel, Perm City, Russia) with mass composition (wt.%) 0.50% Si; 0.50% Fe; 4.20% Cu; 0.70% Mn; 1.50% Mg; Al balance, was used as the substrate material. The substrates had the shape of disks of 45 mm diameter and 15 mm thickness. The coating was applied along the outer diameter (circumference) of each disk. In order to clean and activate the substrate surface, the disks were sandblasted prior to coating deposition with the abrasive particles about 270 μm. The morphology of the surface after sandblasting is shown in Figure 1.

Powder produced by Castolin Eutectic Messer Group (Bad Soden, Germany) was used for the coating deposition. The mass composition of the powder was 10 (wt.%) Co; 4 (wt.%) Cr; WC balance. The particle size distribution of the powder was +10–50 µm. The SEM images of the powder (JEOl, Tokio, Japan) are presented in Figure 2. The powder was produced by agglomerating sintering (Bad Soden, Germany). Each particle contains the tungsten carbide grains with size ≤2 μm embedded in the cobalt and chromium binder.

The detonation spraying was carried out using commercial detonation spraying equipment CCDS2000 (Siberian Technologies of Protective Coatings LLC, Novosibirsk, Russia) presented in Figure 3.

Overall, two batches of samples were produced. During deposition of the first batch of the coatings, the spraying distance was fixed at 270 mm. The second batch was sprayed at a lower distance: 230 mm. The powder feed rate (0.72 g/s) was the same in all cases. The shots were controlled according to a spraying program where the following stages of work were recorded: (1) injection of a combustible mixture of gases into the mixing chamber (60 ms); (2) powder dosage (50–90 ms); (3) closing the valve of the mixing chamber (90–125 ms); (4) ignition of the gas mixture (105 ms); (5) purging the barrel with nitrogen (80–190 ms). The final thickness of the coating was about 300 μm. A mixture of acetylene, propane, and oxygen gases (Chelyabtekhgaz, Chelyabinsk, Russia) was used for the shots. The frequency of shots was 240 ms per shot, that is, approximately 4 shots per second.

Both the outer surface of the coating and the transverse sections of the coatings were analyzed. The surface morphology of the obtained coating and the cross section microstructure of the samples were observed using a JSM-7001F scanning electron microscope (SEM) (JEOL, Tokio, Japan) equipped with an energy dispersive spectrometer from Oxford Instruments (Abingdon, Great Britain) for performing qualitative and quantitative X-ray microanalysis (XRMA). The transverse sections of the samples were also studied to determine the porosity of the coating using an Axio Observer D1.m optical inverted metallographic microscope (Karl Zeiss, Oberkochen, Germany) equipped with the Thixomet Pro software and hardware complex for image analysis. To assess the porosity, four samples were examined for each batch, and at least 10 fields of view were analyzed on each sample.

X-ray phase analysis (XRD) was carried out on the thin sections of the samples on an Ultima IV diffractometer (Rigaku, Japan) using Cu K_α_. The scanning speed was 5 degrees per minute and the scanning step was set to 0.02 degrees.

Microhardness was measured on transverse sections via a FM-800 microhardness meter (Future-Tech Corp., Tokio, Japan). Microhardness measurements of the obtained coating were carried out with a load of 300 g at 30 points according to the following scheme: one parallel of measurement (across the coating) consisted of three dimensions, one such point was at the surface of the coating, the second point—in the central part of the coating, the third point—from the side of the substrate, and there were at least 10 such parallels. Microhardness of the aluminum alloy substrate was measured with a load of 100 g at no less than 20 points, both along and across the substrate.

To perform the coating cohesive strength measurements, the detachable cylindrical samples with threaded heads were manufactured (Figure 4). This sample geometry ensured the retention of the sample in the grips of the testing machine. The total length of the cylindrical sample was 70 mm, the length of the working part was 33 mm, and the diameter of the working part was 5 mm. The connector on the samples excluded the influence of the properties of the base material on the cohesive strength measurements of the coating. The coating was sprayed on the working cylindrical part of the manufactured samples (in the assembly) under the same parameters as for the main samples.

Tests on cohesive strength were carried out in tension on a UTS-110M tensile testing machine (LLC Testsystems, Ivanovo, Russia). In total, three cylindrical samples were prepared to test each application mode.

## 3. Results and Discussion

### 3.1. Morphology and Microstructure

According to an analysis of images of the transverse sections using optical microscope, similar coating porosities are recorded on experimental samples of batch 1 and 2. The value of coating porosity was less than 0.5% (Table 1). One can conclude that the variation of spray distance in the range between 230 and 270 mm does not affect the coating porosity.

The results of studying the morphology and microstructure of the coating in both samples using a scanning electron microscope are shown in Figure 5 and Figure 6. Analysis of the surface morphology of the obtained coating (see Figure 5) showed that on both samples, the coating on the surface is dense, without visible cracks, defects, or large height differences. The size of the pores does not exceed 2 μm. According to the Figure 6a,b, the coating thickness of both samples is comparable and equal to 300 μm. Porosity estimation using SEM images allowed to obtain the porosity value near 0.5% that is close to the results obtain by optical microscopy.

At the point of contact between the substrate (dark gray phase) and the coating (light gray phase), no visible defects or large pores were not observed (Figure 6). The coating is mainly formed by irregularly-shaped particles 0.3–2 μm in size (in some cases the particles are less than 0.1 μm), which can be attributed to tungsten carbide, located in the binding medium, consisting of cobalt and chromium. Comparison of the images of initial powder particles presented in Figure 2b and the deposited coatings allowed to conclude that there the grain size of the tungsten carbide particles is mainly the same in both cases. However, it is important to note that in the applied coating, some amount of WC particles smaller than 0.1 μm were detected. The tungsten carbide particles of similar size were not present in the powder before deposition. In general, the carbide particles are uniformly distributed in the binder, with the exception of individual, single areas of CoCr of sizes of the order of 2–3 μm, where the binder does not have reinforcing particles inside (Figure 6c,d).

The morphology, microstructure, and particle sizes of carbides in the obtained detonation coatings are comparable with the ones of coatings obtained through HVOF [15,16,17,18,19,21]. However, in the detonation spray coating the particle distribution is seems to be more uniform. Similar conclusion was previously made in the work by Roy [29].

The cross-section analysis of etched samples revealed that in the substrate of both batches of samples at the directly under the coating there is a matrix layer with enlarged and partially deformed grains (compared with the rest of the aluminum alloy) with lengths in different areas from 40 to 65 μm (Figure 7). This effect could be explained by significant heat input during coating deposition process. Besides the pulse-periodic regime of spraying and variation of spraying distance, the hot jet of detonation product increases the substrate surface temperature that leads to the grain grow in the substrate. According to Uematsu et al. [18], when using the HVOF process, the aluminum alloy substrate is also subjected to significant thermal input. However, in case of HVOF, the formation of micro cracks, going from the contact zone deep into the aluminum alloy, is often observed. In this study such defects were not observed that indicates on significantly smaller thermal input to the substrate during coating deposition process in comparison with HVOF technique.

### 3.2. Results of XRMA Mapping

The results of mapping using XRMA of the transverse section of specimen 1 are shown in Figure 8. According to the obtained data, the distribution of chromium and cobalt in the binder is uneven; in different areas a clear prevalence of chromium or cobalt is noticeable. According to Bolelli et al. [15] and Thakur et al. [17], part of cobalt can bind to the complex carbide Co_3_W_3_C which could cause a redistribution of elements in the CoCr binder.

### 3.3. XRD Analysis

The results of X-ray phase analysis of the detonating coating for samples 1 and 2 are identical (see Table 1); the X-ray diffraction pattern of the coating on sample 1 is shown in Figure 9. In addition to the presence WC phase small amounts W_2_C and Co_3_W_3_C phases were detected. The possibility of the formation of these phases in a hard cermet coating is also indicated in the works [15,16,17,21,30].

### 3.4. Microhardness HV

The value of coating microhardness was varied from 1126 up to 1725 HV_0.3_ depending on the point of measurement. However, the average value is comparable for both samples (see Table 1). The average microhardness of the substrate was equal to 114 HV_0.1_. It should be noted that, immediately adjacent to the contact zone between the coating and the substrate, the microhardness of the aluminum alloy was somewhat higher and amounted to 133 HV_0.1_, which is consistent with the observed microstructure of substrate.

The microhardness of obtained coating is higher than values reported in previous works, for example, 850–1150 HV_0.1_was presented in the work Bolelli et al. [15] and 775 HV_0.1_ obtained by Roy [30], and was comparable with 1326–1369 HV_0.3_ obtained in the work by Kumari et al. [16], and 1297–1860 HV_0.3_ measured by Thakur et al. [17].

### 3.5. Cohesive Strength

The maximum tensile force values measured during cohesive strength tests for the batch 1 was equal to 584 N, with an average of 573 N. In the case of the second batch sprayed at lower distance, the maximum tensile force at failure was equal to 497 N, with an average equal to 484 N. According to the results of calculations, the average cohesive strength for sample of batch 1 was 236 MPa (see Table 1) whereas for the samples from batch 2 it was 200 MPa (see Table 1), which is comparable with the data obtained by Ulianitsky et al. [31].

## 4. Conclusions

(1)The WC-Co-Cr coatings obtained on aluminum alloy substrates by detonation spraying, with a thickness of 300 μm, at two different spray distances (batch 1 at 270 mm and batch 2 at 230 mm) are dense, without visible defects. The value of coating porosity was less than 0.5% with size of pores not higher than 2 μm.(2)The detonation coating is formed from irregular carbide particles with sizes of the order of 0.3–2 μm that is comparable with the sizes of particles in the initial granules. However, some cases with sizes less than 0.1 μm were found on the carbide fragmentation during spraying. Carbides are evenly distributed in a binder medium consisting of cobalt and chromium.(3)The morphology and the microstructure, as well as the particle sizes of carbides in the obtained detonation coatings are comparable with the literature data on the properties of coatings obtained by the HVOF process, however, the detonation spray coating exceeds the HVOF one in microhardness and density.(4)According to the X-ray phase analysis, the obtained detonation spray coatings mainly consist of the WC phase. Only a negligible amount of W_2_C and Co_3_W_3_C phases were detected.(5)The microhardness values of the carbide coating for both batches samples were approximately the same and amounted to about 1300–1350 HV_0.3_. This value is significantly higher in comparison with tungsten carbide coatings sprayed by various methods on aluminum alloy substrates.(6)According to the obtained results, the variation of spraying distance from 270 to 230 mm does not influence on the coating microstructure and composition. No particular influence on the substrate structure was detected. However, the influence of the distance on the mechanical properties of the coatings was found. In particular, the average cohesive strength of the samples of batch 1 was 236 MPa, whereas for the samples of batch 2 was near 200 MPa. One can conclude that decrease in the spraying distance from 270 to 230 mm diminished the coating cohesive strength.

## Figures and Tables

**Figure 1 materials-14-01206-f001:**
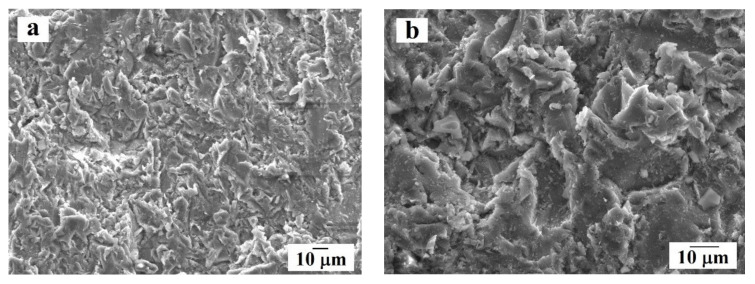
Surface morphology of the aluminum alloy substrate after sandblasting at different magnifications: (**a**) overview; (**b**) enlarged fragment.

**Figure 2 materials-14-01206-f002:**
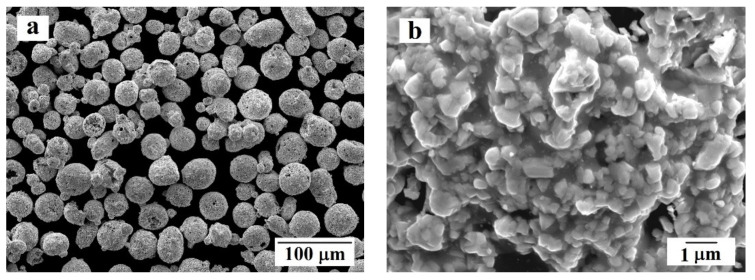
WC-Co-Cr powder used for detonation spray coating deposition: (**a**) overview; (**b**) surface morphology of an individual granule.

**Figure 3 materials-14-01206-f003:**
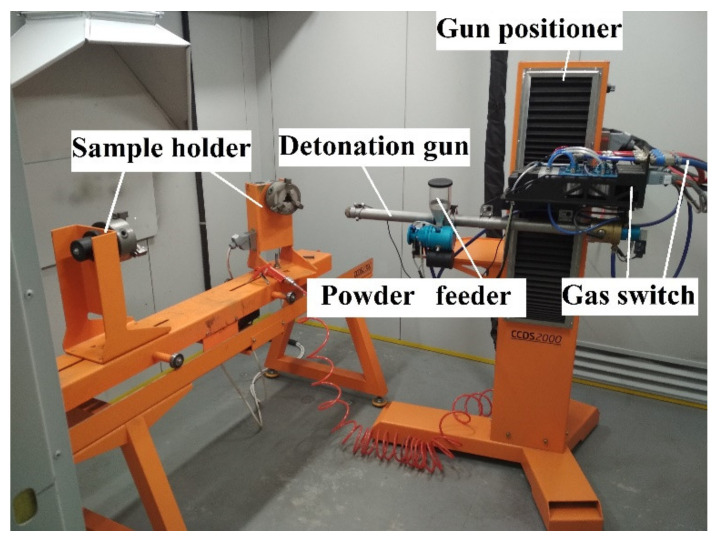
Overview of detonation spray equipment CCDS2000.

**Figure 4 materials-14-01206-f004:**
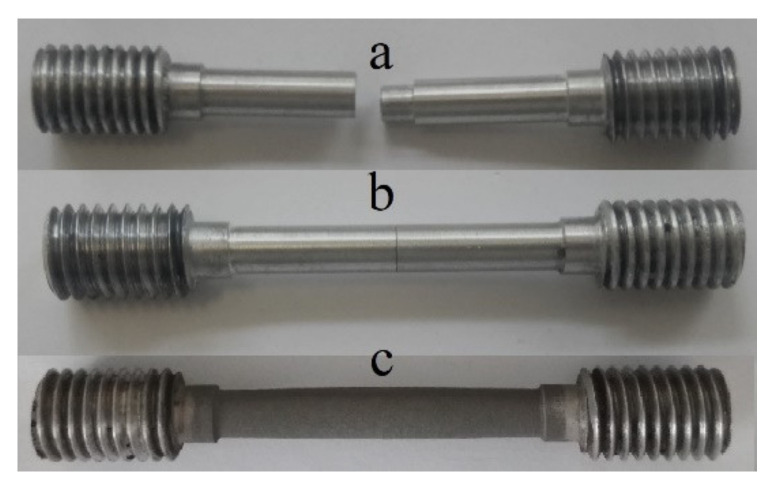
The sample for cohesive strength tests: (**a**) before spraying, disassembled; (**b**) before spraying, assembled; (**c**) after spraying, assembled.

**Figure 5 materials-14-01206-f005:**
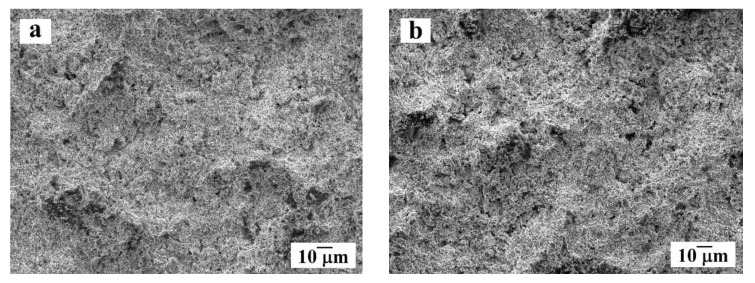
Surface morphology of the coatings: (**a**,**c**,**e**) sample of batch 1; (**b**,**d**,**f**) sample of batch 2.

**Figure 6 materials-14-01206-f006:**
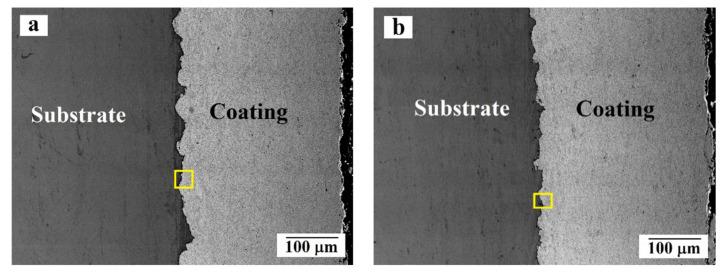
Morphology and microstructure of the cross section of the samples: (**a**,**c**,**e**) sample of batch 1; (**b**,**d**,**f**) sample of batch 2. Enlarged areas in the figures are indicated with yellow squares.

**Figure 7 materials-14-01206-f007:**
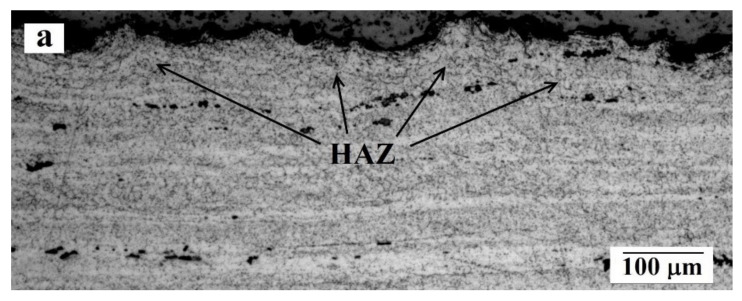
Heat-affected zone (HAZ) in aluminum alloy structure: (**a**) general view; (**b**) enlarged fragment.

**Figure 8 materials-14-01206-f008:**
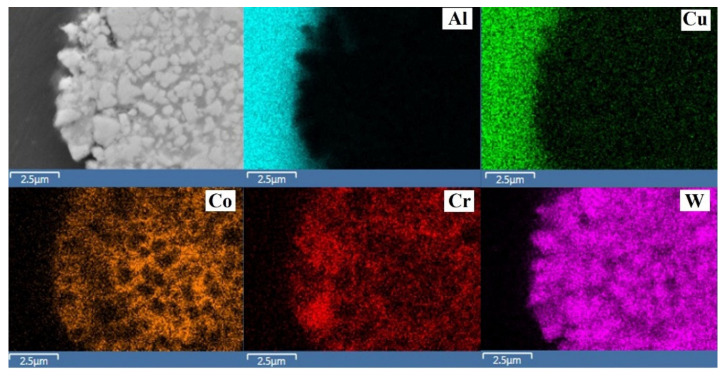
The results of mapping coverage (according to XRMA) on sample of batch 1.

**Figure 9 materials-14-01206-f009:**
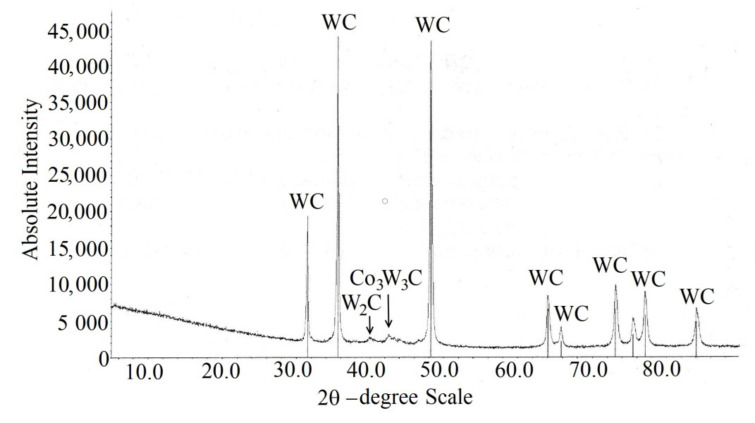
The X-ray diffraction pattern of coating of the sample of batch 1.

**Table 1 materials-14-01206-t001:** Summary of the properties of deposited WC-Co-Cr detonation spray coating in comparison with literature data on the HVOF process.

Batch	Distance to Detonation Gun, mm	Coating Thickness, μm	Porosity, %	Phase Composition *	Hardness, (av.), HV	Cohesion Strength (av.), MPa
1	270	300 ± 5	<0.5	WC	1354 ± 67	236 ± 7
2	230	300 ± 5	<0.5	WC	1315 ± 65	200 ± 6
HVOF [15]	-	50–150	0.4–3.2	WC, W_2_C, Co_3_W_3_C	850–1150 HV_0.1_	-
HVOF [16]	-	220–240	1.0–1.1	WC, W_2_C	1326–1369 HV_0.3_	-
HVOF [17]	-	300	0.57–0.75	WC, W_2_C, Co_3_W_3_C	1297–1860 HV_0.3_	-
HVOF [19]	-	100	0.6	WC, Co	8.55 HV_5_	-
HVOF [21]	-	350	2.0	WC, W_2_C	-	-

* Note: Information on the phase composition is based on a set of data from XRD and XRMA; in addition to the presence WC phase negligible amount (about 5%) of W_2_C and Co_3_W_3_C phases are noted.

## Data Availability

The data presented in this study are available on request from the corresponding author.

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
