# Peer review of "Properties of WC–10%Co–4%Cr Detonation Spray Coating Deposited on the Al–4%Cu–1%Mg Alloy"

_materials, 2021, doi:10.3390/ma14051206_

Round 1
Reviewer 1 Report
In this work the authors characterize the properties of WC–10%Co–4%Cr detonation spray coating deposited on the Al–4%Cu–1%Mg alloy. The research appears to be efficiently done and appropriately reported, however, the the standard of English is acceptable only needs few corrections. Nevertheless, there some questions and corrections that must be answered to improve and complete the document.
Abstract section: The abstract is a little bit confuse and missis some information, I suggest to authors follow these rules:
- One or two sentences on BACKGROUND
- Two or three sentences on METHODS
- Less than two sentences on RESULTS
- One sentence on CONCLUSIONS
Introduction section: In this section the authors don’t indicate the novelty of their work. what is the innovation of your work when compared with the other researchers? The "Knowledge gap to be filled"? In this introduction the authors must describe or indicate the work that will be done to test their "hypothesis".
Lines 40, 4332, 56, 56. The authors indicate the references only by the number. However, when they want referrer a particular researcher(s) the authors must indicate his/her/their name(s). Example of line 40: According to Nenashev et. al [16]. This mistake is repeated in other parts of the manuscript.
Lines 120-123. The authors describe the microhardness measurements. However, they did not indicate where these measurements were done on the specimen. Please represent schematically a representation of the specimen measurement field, such as, the specimen region of the measurements
Line 175. Please, change “[15] However …” to “[15]. However …”
Figure 6. Please, please indicate, as zoom circle, the regions where are done the successive analyses (ex. a, c, e).
Author Response
Dear Reviewer! First of all, we would like to express our deep gratitude to you for your attentive attitude to our work. Thank you for your valuable comments that have made our manuscript better. Below we provide answers to your comments. Also, the changes made to the text of the manuscript according to your comments are highlighted in yellow marker.
Comment 1. Abstract section: The abstract is a little bit confuse and missis some information, I suggest to authors follow these rules:
- One or two sentences on BACKGROUND
- Two or three sentences on METHODS
- Less than two sentences on RESULTS
- One sentence on CONCLUSIONS
Answer: Corrections have been made to the text of the manuscript:
Abstract: One of the method of local improvement of the wear resistance of aluminum alloy parts is the deposition of hard tungsten carbide-based coatings on the surfaces subjected to intense external influence. This paper is devoted to the characterization of the WC–10%Co–4%Cr coating deposited on an Al–4%Cu–1%Mg alloy by the detonation spray method. In comparison with the common thermal spray techniques like HVOF (High Velocity Oxygen Fuel) or APS (Arc Plasma Spraying), the heat input delivered to the substrate during detonation spray is significantly lower, that is especially important in case of coating deposition on aluminum alloys. The paper presents the results of morphology investigation, microstructure, phase composition, microhardness, and cohesive strength of deposited carbide-based detonation spray coating. Results showed that the coating has a porosity less than 0.5 % and the carbide grain refinement down to the submicron size during coating deposition was detected. According to the investigation, the variation of spraying distance from 270 mm to 230 does not influence on the coating microstructure and composition.
Comment 2. Introduction section: In this section the authors don’t indicate the novelty of their work. what is the innovation of your work when compared with the other researchers? The "Knowledge gap to be filled"? In this introduction the authors must describe or indicate the work that will be done to test their "hypothesis".
Answer: Corrections have been made to the text of the manuscript:
It is important to note that the results presented in literature described mainly the properties of tungsten carbide coatings with WC phase percentage in feedstock powder not higher than 75%. In case of deposition of cermet powders with higher carbide percentage (with WC content up to 86–88%) significantly higher energy explosion energy should be used in order to obtain dense and uniform coating. However, increase of the explosion energy could lead to undesirable effects related to the overheating of aluminum surface with formation of new phases on the coating/substrate interface. Unfortunately, the data presented in literature do not fully describe the properties of detonation spray coatings with WC content up to 86–88% deposited on the aluminum alloy substrate.
In the other hand, in comparison with the common thermal spray techniques like HVOF (High Velocity Oxygen Fuel) or APS (Arc Plasma Spraying), the heat input delivered to the substrate during detonation spray is significantly lower, that is especially important in case of coating deposition on aluminum alloys.
In our opinion, the application of high-carbide coatings (with WC content up to 86–88%) on aluminum alloys by the detonation method should not lead to a deterioration in the quality of the resulting coating, which we intend to show in our study.
Comment 3. Lines 40, 4332, 56, 56. The authors indicate the references only by the number. However, when they want referrer a particular researcher(s) the authors must indicate his/her/their name(s). Example of line 40: According to Nenashev et. al [16]. This mistake is repeated in other parts of the manuscript.
Answer: Corrections have been made to the text of the manuscript.
Comment 4. Lines 120-123. The authors describe the microhardness measurements. However, they did not indicate where these measurements were done on the specimen. Please represent schematically a representation of the specimen measurement field, such as, the specimen region of the measurements
Answer: Corrections have been made to the text of the manuscript:
Microhardness was measured on transverse sections via a FM-800 microhardness meter. Microhardness measurements the obtained coating were carried out with a load of 300 g at not less than thirty points according to the following scheme: one parallel of measurement (across the coating) consisted of three dimensions, such as one point was at the surface of the coating, the second point - in the central part of the coating, the third point - from the side of the substrate, and such parallels were at least ten. Microhardness of aluminum alloy substrate was measured with a load of 100 g at no less than twenty points both along and across the substrate.
Comment 5. Line 175. Please, change “[15] However …” to “[15]. However …”
Answer: Corrections have been made to the text of the manuscript.
Comment 6. Figure 6. Please, please indicate, as zoom circle, the regions where are done the successive analyses (ex. a, c, e).
Answer: The figure has been corrected according to the comment.

Reviewer 2 Report
I read this article with interest because it covers the relatively difficult issue of making very hard coatings on a low-melting and soft substrate.
The obtained test results concerning the mechanical properties do not raise any doubts and are presented correctly.
I only have few doubts/ question / notice to discuss
First about the scale in Figure 6. It seems to me that the scale on the photo in Figure c and d or e and f is wrong, i.e. in e and f there are carbides much larger than in c and d. The size of carbides presented in Figure 2b is definitely smaller than in layer. Moreover, on line 159-160 it is indicated that the size of the carbides is the same. I think it is worth verifying.
In line 177 it is indicated that the microstructure of the aluminum was examined and that etching was performed for evaluation. There was no such indication in the research methodology. It seems to me that it is worth supplementing this information, as well as showing photographs of the structure of the starting material and HAZ. Especially that line 185 indicated the presence of microcracks in the area of HAZ in the substrate material made by HVOF, which was not revealed (not observed) in your research.
Looking at Figures 6c and 6d, it seems to me that there are cracks in the coating in areas with lower carbide density. Moreover, in Figure 6b, thin cracked layers are also visible near to the surface.
A very high hardness of the coating and a relatively high strength of substrate-coating cohesion were obtained. However, the obtained values are lower than the yield strength of the Al-Cu-Mg alloys, with high thermal expansion coefficient of the alloy at the same time. A question arises here whether the coating will not peel or transverse cracks due to the heating of the substrate? I think that it is worth comparing this problem with conclusion 5 because while the aim of the research was to obtain the highest possible hardness of the coating, the performance properties of the joint itself may be very low.
Author Response
Dear Reviewer! First of all, we would like to express our deep gratitude to you for your attentive attitude to our work. Thank you for your valuable comments that have made our manuscript better. Below we provide answers to your comments. Also, the changes made to the text of the manuscript according to your comments are highlighted in green marker.
Comment 1. First about the scale in Figure 6. It seems to me that the scale on the photo in Figure c and d or e and f is wrong, i.e. in e and f there are carbides much larger than in c and d. The size of carbides presented in Figure 2b is definitely smaller than in layer. Moreover, on line 159-160 it is indicated that the size of the carbides is the same. I think it is worth verifying.
Answer: Corresponding corrections were made in Figure 6.
Comment 2. In line 177 it is indicated that the microstructure of the aluminum was examined and that etching was performed for evaluation. There was no such indication in the research methodology. It seems to me that it is worth supplementing this information, as well as showing photographs of the structure of the starting material and HAZ. Especially that line 185 indicated the presence of microcracks in the area of HAZ in the substrate material made by HVOF, which was not revealed (not observed) in your research.
Answer: we have added Figure 7.
Comment 2. Looking at Figures 6c and 6d, it seems to me that there are cracks in the coating in areas with lower carbide density. Moreover, in Figure 6b, thin cracked layers are also visible near to the surface.
Answer: These small artifacts in the figures are the result of etching the substrate. Unfortunately, washing the sample did not remove all contamination.
Comment 3. A very high hardness of the coating and a relatively high strength of substrate-coating cohesion were obtained. However, the obtained values are lower than the yield strength of the Al-Cu-Mg alloys, with high thermal expansion coefficient of the alloy at the same time. A question arises here whether the coating will not peel or transverse cracks due to the heating of the substrate? I think that it is worth comparing this problem with conclusion 5 because while the aim of the research was to obtain the highest possible hardness of the coating, the performance properties of the joint itself may be very low.
Answer: We plan to study changes in performance over time, but these materials science and trebological tests are not included in this work. Thank you so much for this comment, now we know exactly what to focus on in further research.

Reviewer 3 Report
The reviewed article deals with WC-Co-Cr coatings manufactured by detonation spraying and its properties. In general the whole paper is well organized. Nevertheless, there are some remarks which should be pointed out:
- Line 18 - please use "Atmospheric Plasma Spraying" instead "Arc Plasma Spraying".
- What was the surface roughness after sand-blasting?
- What was a d50 of the feedstock material?
- Please add the informations: (i) which gases were used and (ii) what was the frequency of "shots"?
- How many images were taken into accout in order to estimate coatings' porosity?
- In XRD investigations please add: (i) what was a scan step and (ii) time for step?
- Why different values of the maximum load (for coating and substrate) were used during microhardness measurements?
- Please add or comment why there is no XRMA and XRD results of sample no. 2. Moreover, in Table 1 there is no information how much of WC phase is sample 1 and sample 2.
- Maybe an image of fracture section after cohesive strength test should be added and discussed?
- Table 1 - please add for thickness, hardness and cohesion strength also standard deviation.
- For the introduction part please add also:
https://doi.org/10.3390/coatings11020218
Author Response
Dear Reviewer! First of all, we would like to express our deep gratitude to you for your attentive attitude to our work. Thank you for your valuable comments that have made our manuscript better. Below we provide answers to your comments. Also, the changes made to the text of the manuscript according to your comments are highlighted in red marker.
Comment 1. Line 18 - please use "Atmospheric Plasma Spraying" instead "Arc Plasma Spraying".
Answer: the corresponding corrections have been made in the text of the manuscript.
Comment 2. What was the surface roughness after sand-blasting?
Answer: the corresponding corrections have been made in the text of the manuscript:
In order to clean and activate the substrate surface, the disks were sandblasted prior to coating deposition with the abrasive particles about 270 μm.
Comment 3. What was a d50 of the feedstock material?
Answer: The particle size distribution of the powder was +10-50 µm.
Comment 4. Please add the informations: (i) which gases were used and (ii) what was the frequency of "shots"?
Answer: the corresponding corrections have been made in the text of the manuscript:
A mixture of acetylene, propane and oxygen gases was used for the shots. The frequency of shots was 240 ms per shot, that is, approximately 4 shots per second.
Comment 5. How many images were taken into accout in order to estimate coatings' porosity?
Answer: the corresponding corrections have been made in the text of the manuscript:
To assess the porosity, four samples were examined for each batch, at least ten fields of view were analyzed on each sample.
Comment 6. In XRD investigations please add: (i) what was a scan step and (ii) time for step?
Answer: the corresponding corrections have been made in the text of the manuscript:
The scanning speed was 5 degrees per minute and the scanning step was set to 0.02 degrees.
Comment 7. Why different values of the maximum load (for coating and substrate) were used during microhardness measurements?
Answer: Different loads were used due to the apparent difference in hardness between the substrate and the coating. And the load was selected based on the need to accurately determine the size of the indenter print.
Comment 8. Please add or comment why there is no XRMA and XRD results of sample no. 2. Moreover, in Table 1 there is no information how much of WC phase is sample 1 and sample 2.
Answer: The data for sample number 2 are not presented, since they are almost identical to sample number 1. Data on the quantitative phase composition have been added to the text and highlighted with a red marker.
Comment 9. Maybe an image of fracture section after cohesive strength test should be added and discussed?
Answer: unfortunately, after the tests, the fracture surfaces in the coating area were not suitable for fractographic studies.
Comment 10. Table 1 - please add for thickness, hardness and cohesion strength also standard deviation.
Answer: the corresponding corrections have been made in the text of the manuscript.
Comment 11. For the introduction part please add also:
https://doi.org/10.3390/coatings11020218
Answer: Thank you, we added this article.

Reviewer 4 Report
Thank you for submitting this work to materials. I think this is a good fit and I hope we can publish this soon.
Below a few comments I would like you to consider as I think that they can further improve the quality of your work.
Abstract:
-significant lower - by how much for your WC-Cr particles? Can you quantify?
Introduction:
-maybe mention that it is a discontinuous process and therefore the heat input is much lower
-maybe include a figure or table where you compare the heat input, or particle acceleration, or other characteristic of the different coating methods. Another interesting question would be the amount of particles that you can depost with the different methods.
Materials and Methods:
-what is the reason for the different distances 230 and 270 mm - please add one sentence to elaborate
-how much was the powder feed rate
-what gases were used
Results and Discussion:
-Figure 5 - include the differrent production parameters in the pictures in Fig. 5
-Fig. 6 - again include parameters if possible
-You compare the structures a lot with those obtained using HVOF - do you have hardness data etc. for similar HVOF samples? I think a direct comparison between the two methods would be very very interesting.
Conclusions:
-You should not present new results here - No. 3 seems to be new to me. Include a table, or figure where you compare the hardness of the different methods in the result section and you should be fine.
What is your recommendation 230 or 270 mm distance? Does it really matter? and why?
__________________
Great work, please include the recommendations and I am confident we can publish this quickly.
Author Response
Dear Reviewer! First of all, we would like to express our deep gratitude to you for your attentive attitude to our work. Thank you for your valuable comments that have made our manuscript better. Below we provide answers to your comments. Also, the changes made to the text of the manuscript according to your comments are highlighted in blue marker.
Comment 1. Abstract:
-significant lower - by how much for your WC-Cr particles? Can you quantify?
Answer: The calculated part is in the process, and we are not ready to present it yet, so as not to mislead. But, that the heat transfer is below can be said with confidence.
Comment 2. Introduction:
-maybe mention that it is a discontinuous process and therefore the heat input is much lower
-maybe include a figure or table where you compare the heat input, or particle acceleration, or other characteristic of the different coating methods. Another interesting question would be the amount of particles that you can depost with the different methods.
Answer: At the moment we are comparing our detonation spraying with the literature data on HVOF, since the installation for HVOF is in the process of purchasing by our university. In the future, we plan to expand our research.
Comment 3. Materials and Methods:
-what is the reason for the different distances 230 and 270 mm - please add one sentence to elaborate
Answer: Different distances were chosen in accordance with the reasoning that the distance from the gun to the sample of 270 mm is standard, but we wanted to test our hypothesis that reducing the distance can favorably affect the quality of the formed surface.
Comment 4. -how much was the powder feed rate
Answer: Corresponding corrections have been added to the text of the manuscript:
The powder feed rate was 0.72 g/s.//
Comment 5. -what gases were used
Answer: Gas composition has been added to the text.
Comment 6. Results and Discussion:
-Figure 5 - include the differrent production parameters in the pictures in Fig. 5
-Fig. 6 - again include parameters if possible.
Answer: We tried to fix the drawings, but as a result they became much worse and more unreadable. Sorry, we left the original version.
Comment 7. -You compare the structures a lot with those obtained using HVOF - do you have hardness data etc. for similar HVOF samples? I think a direct comparison between the two methods would be very very interesting.
Answer: At the moment we are comparing our detonation spraying with the literature data on HVOF, since the installation for HVOF is in the process of purchasing by our university. In the future, we plan to expand our research.
Comment 8. Conclusions:
-You should not present new results here - No. 3 seems to be new to me. Include a table, or figure where you compare the hardness of the different methods in the result section and you should be fine.
Answer: We have supplemented the table with literature data on the HVOF process.
Comment 9. What is your recommendation 230 or 270 mm distance? Does it really matter? and why?
Answer: The fact that the quality of the coating practically did not change was unexpected. Our further studies will be associated with an even greater decrease in the distance from the gun to the sample surface.

Round 2
Reviewer 1 Report
The second version of the manuscript improved significantly when compared with the first version. So, in my opinion, the manuscript can be accepted for publication.
Author Response
Dear Reviewer.
We would like to thank you once again for your attentive attitude towards our work and very constructive comments.
We agree that our work has improved significantly.
Reviewer 3 Report
All my remarks have been included.
Author Response

(The authors gave the same response as above.)

Reviewer 4 Report
Excellent - this can be published now
Author Response
Dear reviewer.
We would like to thank you once again for your attentive attitude towards our work and very constructive comments.
We agree that our work has improved significantly.